

# Phylogenomics and classification of *Notropis* and related shiners (Cypriniformes: Leuciscidae) and the utility of exon capture on lower taxonomic groups

Carla Stout[1], Susana Schonhuth[2], Richard Mayden[2], Nicole L. Garrison[3] and Jonathan W. Armbruster[4]

[1] Department of Biological Sciences, California State Polytechnic University, Pomona, Pomona, CA, United States of America
[2] Department of Biology, Saint Louis University, St. Louis, MO, United States of America
[3] Department of Biology, West Liberty University, West Liberty, WV, United States of America
[4] Department of Biological Sciences, Auburn University, Auburn, AL, United States of America

Corresponding author
Jonathan W. Armbruster,
armbrjw@auburn.edu

## ABSTRACT

North American minnows of the Shiner Clade, within the family Leuciscidae, represent one of the most taxonomically complex clades of the order Cypriniformes due to the large number of taxa coupled with conserved morphologies. Species within this clade were moved between genera and subgenera until the community decided to lump many of the unclassified taxa with similar morphologies into one genus, *Notropis*, which has held up to 325 species. Despite phylogentic studies that began to re-elevate some genera merged into *Notropis*, such as *Cyprinella*, *Luxilus*, *Lythrurus*, and *Pteronotropis*, the large genus *Notropis* remained as a taxonomic repository for many shiners of uncertain placement. Recent molecular advances in sequencing technologies have provided the opportunity to re-examine the Shiner Clade using phylogenomic markers. Using a fish probe kit, we sequenced 90 specimens in 87 species representing 16 genera included in the Shiner Clade, with a resulting dataset of 1,004 loci and 286,455 base pairs. Despite the large dataset, only 32,349 bp (11.29%) were phylogenetically informative. In our maximum likelihood tree, 78% of nodes are 100% bootstrap supported demonstrating the utility of the phylogenomic markers at lower taxonomic levels. Unsurprisingly, species within *Notropis* as well as *Hudsonius*, *Luxilus*, and *Alburnops* are not resolved as monophyletic groups. *Cyprinella* is monophyletic if *Cyprinella callistia* is excluded, and *Pteronotropis* is monophyletic if it includes *Hudsonius cummingsae*. Taxonomic changes we propose are: restriction of species included in *Alburnops* and *Notropis*, elevation of the subgenus *Hydrophlox*, expansion of species included in *Miniellus*, movement of *Hudsonius cummingsae* to *Pteronotropis*, and resurrection of the genera *Coccotis* and *Paranotropis*. We additionally had two specimens of three species, *Notropis atherinoides*, *Ericymba amplamala*, and *Pimephales vigilax* and found signficant differences between the localities (1,086, 1,424, and 845 nucleotides respectively).

## INTRODUCTION

Among North American cyprinoids, the shiners and related minnows have been among the most taxonomically complex groups of fishes. The group is currently placed in the Leuciscidae, subfamily Pogonichthyinae after the elevation of subfamilies to family rank in cyprinoids (*Schönhuth et al., 2018*; *Tan & Armbruster, 2018*). Ichthyologists tasked with assembling species into meaningful genera initially described a dizzying array of genera and subgenera of minnows. Species were moved between various categories (genera, subgenera, tribes) based primarily on phenetic similarity; many leuciscid genera remained relatively stable but one, *Notropis* (and many taxa formally placed in this genus), has continued to be difficult. A search of the Catalog of Fishes (*Fricke, Eschmeyer & van der Laan, 2022*) for '*Notropis*' results in 325 species, representing a considerable bulk of nominal species of North American freshwater fishes.

Starting with *Mayden (1989)* comprehensive morphological phylogenetic analysis of *Cyprinella* and other North American taxa, a large monophyletic group was recognized as the Open Posterior Myodome clade (OPM). This clade includes most of the eastern North American leuciscids including *Notropis* and related genera. *Mayden (1989)* concluded that *Notropis* is an artificial group due to convergence of morphological characters and classification by phenetic similarity, and enacted nomenclatural changes elevating some subgenera in *Notropis*, such as *Cyprinella, Luxilus, Lythrurus*, and *Pteronotropis*, recognized other genera and reallocated species. He moved some species from *Notropis* to *Hybopsis* (including *N. boucardi, N. calientis, N. dorsalis, N. longirostris, N. sabinae, N. alborus, N. bifrenatus*), and differentiated six species of *Notropis* (*i.e. N. topeka, N. mekistocolas, N. atrocaudalis, N. stramineus, N. chihuahua* and *N. procne*) from the large clade that included all of the genera previously included in *Notropis*. A few now recognized genera were then not recognized (*i.e. Codoma* was within *Cyprinella*; *Opsopoedus* was within *Notropis*). Still, *Notropis* remained as a "taxonomic repository for small, silvery fishes of unknown relationship" (*Gidmark & Simons, 2014*: 379; see also *Mayden et al., 2006*) with approximately 91 currently recognized species loosely organized into subgenera (*Jordan, 1885*). Primarily because of the large number of taxa, coupled with conserved morphologies, few studies have attempted to tackle the remaining species allocated to the genus or other orphaned taxa of unknown taxonomic placement (*Mayden et al., 2006*; *Hollingsworth et al., 2013*; *Schönhuth et al., 2018*). However, even when necessary taxonomic decisions for species within the genus *Notropis* were made (*sensu Mayden et al., 2006*; *Gidmark & Simons, 2014*), they have not been widely accepted, and traditional taxonomic groups have been preferred by some for nomenclatural stability until a stronger consensus is reached on proposed nomenclatorial changes (*Fricke, Eschmeyer & van der Laan, 2022*).

Phylogenetically, most previous studies on shiners and relatives have focused on resolving relationships within purported monophyletic subgenera (for example *Snelson,*

*1972; Buth, 1979; Raley, Wood & McEachran, 2001; Cashner, Piller & Bart, 2011*) with varied results, and without investigation into relationships among the subgenera or to genera that have been segregated from *Notropis*. *Mayden et al. (2006)* attempted a comprehensive study of the so-called Notropin clade (the clade name does not refer to a taxonomic rank) to resolve relationships using cytb (mitochondrial marker) and revealed a nonmonophyletic *Notropis* as reported in prior studies either based on morphological or molecular data (*Mayden, 1989; Simons, Berendzen & Mayden, 2003*). The former study, in addition to corroborating monophyly of genera synonymized with *Notropis* as *Cyprinella*, *Erycymba, Hybognathus, Hybopsis, Lythrurus*, also identified a more restricted *Notropis* by recognizing the additional genera *Alburnops, Aztecula, Graodus, Hudsonius, Miniellus*, and *Yuriria*. These authors could not resolve *Hydrophlox, Luxilus* and *Pteronotropis* as monophyletic, and identified a 'Notropis' longirostris clade that included seven species of *Notropis*. Despite all these monophyletic groups having available names, their compositions were not always as initially proposed, and relocation of some species were made by *Mayden (1989), Coburn & Cavender (1992)*, and *Mayden et al. (2006)*. However, following this alternative phylogenetic classification of Notropin shiners, many species remained relegated to 'Notropis' (in single quotes) because of their uncertain placement due to weak support in analyses, and relationships among the genera listed above remained unclear. While these additional genera from within a nonmonophyletic *Notropis* were recognized and elevated, most subsequent studies reverted back to a larger encompassing *Notropis*, with perhaps recognition of some of these genera as subgenera or species groups (*Bird & Hernandez, 2007; Rüber et al., 2007; Zhang et al., 2008; Chen & Mayden, 2009; Fang et al., 2009; Gaubert, Denys & Oberdorff, 2009; Scott et al., 2009; Bufalino & Mayden, 2010; Houston, Shiozawa & Riddle, 2010; Cashner, Piller & Bart, 2011; Wang et al., 2012; Hollingsworth et al., 2013; Imoto et al., 2013; Fricke, Eschmeyer & van der Laan, 2022*).

   *Hollingsworth et al. (2013)* expanded upon the cytb study by adding the RAG1 (nuclear) molecular marker for a phylogenetic reconstruction to test for a correlation between a shift from benthic to pelagic lifestyles and increased diversification rates. Unsurprisingly, this analysis also resulted in a relatively poorly resolved overall phylogeny with moderate support for non-monophyly of *Notropis*, but again illustrated the importance of understanding relationships to better inform our understanding of ecological and evolutionary processes.

   To promote further study into this group, *Gidmark & Simons (2014)* amassed much of the knowledge reported for the shiners (distributions, histories, ecologies, *etc.*) and proposed using the designations made by *Mayden et al. (2006)* with the understanding that the relationships among them still remain unclear, despite support for the Shiner Clade as a whole (*Simons, Berendzen & Mayden, 2003; Mayden et al., 2006; Schönhuth et al., 2008*). Recently, a classification of the Holarctic family Leuciscidae based on nuclear and mitochondrial genes was proposed, where the Shiner Clade was a well-supported group within the Pogonichthyinae (*Schönhuth et al., 2018*). Despite the increase in both taxon and character sampling, *Notropis* was not resolved as monophyletic, as species of this genus were found in different parts of the Shiner Clade, and some of the genera formerly included in *Notropis* were not supported as monophyletic (*Alburnops, Notropis s.s.*,

*Hudsonius*), or their composition differed slightly (*i.e. Miniellus, Hydrophlox*) from that previously proposed (*Coburn, 1982; Mayden et al., 2006; Cashner, Piller & Bart, 2011*).

Recent advances in sequencing technologies have provided the opportunity to re-examine the shiner clade using phylogenomic markers. Most phylogenomic-scale studies thus far have focused on higher taxonomic levels (*Lemmon, Emme & Lemmon, 2012; Bond et al., 2014; Eytan et al., 2015; Prum et al., 2015; Hamilton et al., 2016*), but decreases in costs and the establishment of universal loci specific to fishes (*Betancur-R et al., 2013; Arcila et al., 2017; Hughes et al., 2018, 2021*) have helped overcome the hurdles associated with applying a phylogenomic approach to the shiner clade. In this study, we employ the probes developed by *Arcila et al. (2017)* in an attempt to tackle the systematic problems of *Notropis* and related genera, and follow the taxonomy discussed in *Gidmark & Simons (2014)*. We also examined two specimens of three species from different geographic locations to test the utility of the markers on a smaller geographic scale. Previous research (*Rincon-Sandoval, Betancur-R & Maldonado-Ocampo, 2019*) had shown good utility at population-level scales, but wih the shiners, we wanted to determine if the markers could lead to a stong phylogenetic hypothesis for a group with very rapid divergence (*Hollingsworth et al., 2013; Burress et al., 2017*).

## MATERIALS AND METHODS

### Taxon selection, tissue preparation, and sequencing

Every effort was made to acquire broad representation across the shiner genera. For the present study we included 88 ingroup taxa from 16 genera currently included in the Shiner Clade as proposed by *Schönhuth et al. (2018)* as well as outgroup taxa from five different genera (*Notemigonus, Chrosomus, Erimystax, Phoxinus* and *Semolitus*; Table S1 shows genera with number of recognized species, type species, and species sampled). Six genera from the Shiner Clade were not available for this study including *Tampichthys* (six species), *Yuriria* (three species), *Algansea* (seven species), *Aztecula* (two species), *Agosia* (one species), and *Erimonax* (one species). Except for *Agosia* and *Erimonax* all other unsampled genera are endemic to Mexico. To test the utility of the markers at an even smaller taxonomic scale, we include two specimens each of *Notropis atherinoides, Ericymba amplamala*, and *Pimephales vigilax*.

DNA was extracted from specimens using the Omegabiotek E.Z.*N.A.* animal tissue extraction kit (product #D3396-02) following manufacturer protocols. Extracted DNA was checked for quality using electrophoresis and quantity using nanodrop. After ensuring high molecular weight and a minimum of 2 μg total DNA, samples were sent for library preparation and Illumina sequencing to MYcroarray (now Arbor Biosciences, arborbiosci. com). Probes developed by *Arcila et al. (2017)* were used to target 1,060 loci. GenBank project number is PRJNA842507; aligment (File S1) and partition file (File S2) are provided.

### Bioinformatics and tree reconstruction

FASTQ files were uploaded to the Alabama Supercomputer Center (ASC) for preliminary quality control processing. Trimmomatic (*Bolger, Lohse & Usadel, 2014*) was used to
remove adapters and remove leading and trailing low quality bases in the paired end reads, as well as to remove reads with a length less than 36 base pairs. Resulting reads were then imported into Geneious v 6.1.8 (www.geneious.com), set as paired reads, and assembled using the zebrafish (*Danio rerio*) reference for the concatenated loci using five iterations and trimmed to each reference locus. The loci for each species were then concatenated and all concatenations were aligned in Geneious v 6.1.8 with the native alignment tool (www.geneious.com). Tree reconstruction was performed on the Center for Advanced Science Innovation and Commerce (CASIC) computer cluster at Auburn University, Auburn, AL, USA. RAxML (Randomized Axelerated Maximum Likelihood, v. 8.0.24, *Stamatakis, 2014*) was implemented using GTR + G model of evolution on the partitioned loci (partitioned per *Arcila et al., 2017*) and the resulting tree then subjected to 500 bootstrap replicates (BS is percent of trees showing this result). Species tree reconstruction was conducted using ASTRAL-II (*Mirarab & Warnow, 2015*) on individual RAxML gene trees that were subjected to 100 bootstrap replicates. Approximately-unbiased (AU) tests were conducted using CONSEL v.0.20 (*Shimodaira, 2002*) to specifically test the unconstrained maximum likelihood best tree topology against trees that were constrained to force monophyly for three genera: *Cyprinella*, *Hudsonius*, and *Luxilus*. Number of phylogenetically informative sites was calculated in R (v. 4.0.2; *R Core Team, 2020*) using the pis command in the ips package (Interfaces to Phylogenetic Software in R; *Heibl, Cusimano & Krah, 2014*).

## RESULTS

The final alignment yielded 1,004 loci, 286,455 base pairs, and only 0.42% missing data. Of those sites, 32,349 (11.29%) were phylogenetically informative. The range of locus size was 196–1,748 bp, with an average bp length of 285 (Fig. 1). In the resulting concatenated ML tree, 78% of nodes are 100% bootstrap supported with only six nodes collapsing below the 70% bootstrap threshold (Fig. 2). Species tree analysis produced highly congruent results, particularly at the genus level (Fig. 3). At deeper nodes there is less support in the species tree for the placement of a few clades (*i.e. Hudsonius hudsonius + H. altipinnis*; '*Notropis*' *atrocaudalis* + '*N.*' *bifrenatus* + '*N.*' *heterolepis*), resulting in remaining uncertainty as to the relationships among the genera. Nevertheless, with our focus on resolving within-genera relationships, both concatenation and species tree approaches resolve the same patterns with strong support.

Species that are currently placed in *Notropis* are not resolved as monophyletic (Figs. 2 and 3). *Notropis jemezanus*, *N. amabilis*, *N. micropteryx*, *N. rubellus*, and *N. amoenus* form a clade with the type species, *N. atherinoides*, and a second clade included *N. buchanani*, *N. wickliffi*, *N. volucellus*, and *N. spectrunculus*. Species designated as '*Notropis*' by *Mayden et al. (2006)* are found throughout the tree. The genera *Hudsonius*, *Luxilus*, and *Alburnops* were not monophyletic. *Cyprinella* forms a monophyletic group that excludes *C. callistia*, which forms a trichotomy with *Opsopoeodus* + *Pimephales* and the clade containing the remaining members of *Cyprinella* + *Codoma*. The results of the AU test were all significant (*Cyprinella* constrained, $p = 3e{-}06$; *Hudsonius* constrained, $p = 6e{-}08$; *Luxilus* constrained, $p = 2e{-}18$), indicating that all of the constrained topologies can be rejected as alternative tree hypotheses. For comparison, a summary of the four-gene phylogeny of
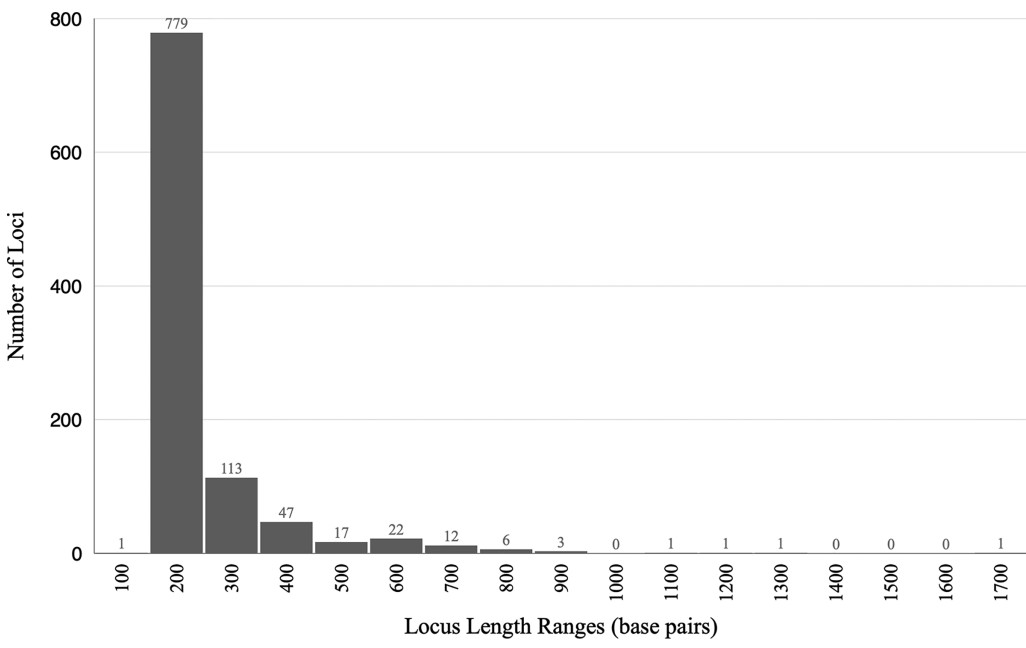

**Figure 1 Histogram showing lengths of loci in base pairs.** The range of locus size was 196–1,748 bp, with an average bp length of 285.

*Schönhuth et al. (2018)* for the shiner clade is presented including the taxonomic changes proposed herein (Fig. 4). A list of all species, genera, and proposed taxonomic changes discussed below are given in Table S2.

## DISCUSSION

Alternative taxonomic and systematic classifications have been proposed for species included within the controversial genus *Notropis* and various purported relatives in several studies based on different characters (*Swift, 1970*; *Coburn, 1982*; *Mayden, 1989*; *Bielawski & Gold, 2001*; *Mayden et al., 2006*; *Gidmark & Simons, 2014*; *Schönhuth et al., 2018*). However, while several genera have been elevated from synonymy with *Notropis* and are monophyletic groups, consensus regarding number and composition of some of the different clades and groups of the genus *Notropis* has remained elusive, and classifications were considered provisional until a more comprehensive study including sufficient taxon and character sampling can produce a well supported analysis of relationships (*Cashner, Piller & Bart, 2011*; *Gidmark & Simons, 2014*; *Schönhuth et al., 2018*). This study provides the results on which to make some, but not all of the remaining taxonomic considerations (Table S2). In general, we made taxonomic changes if this study and that of *Schönhuth et al. (2018)* concur and if there was high support values in Figs. 2 and 3 (>93% bootstrap and/or a local posterior probability of 1 in the ASTRAL-II tree). Taxonomic discussion roughly follows Fig. 2. Boostrap support (BS, Fig. 2) and local posterior probability (PP, Fig. 3) are indicated in the discussion of clades.

All phylogenomic analyses (based on concatenation and species tree approaches) resolved the same patterns with strong support. The 42 species analyzed previously included in *Notropis* were not resolved as monophyletic; however, analyses resolved these
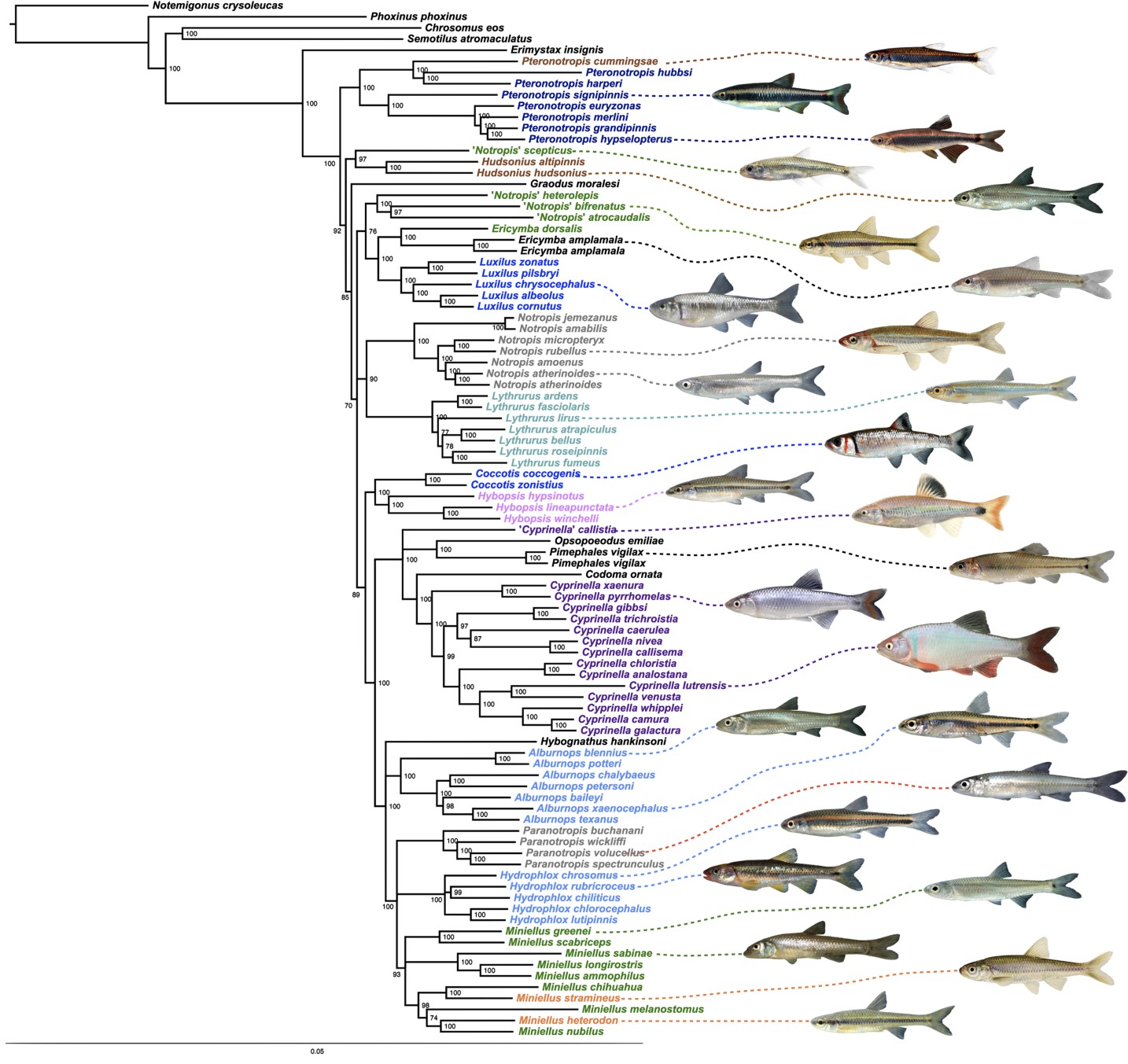

**Figure 2 ML tree based on concatenated alignment.** Numbers of nodes represent bootstrap support, with nodes less than 70% supported collapsed. Scale bar represents number of substitutions per site.

species in several well-supported clades, and nine other well-recognized genera, together providing an alternative classification for this controversial group of shiners. This revisionary classification for *Notropis* supported the recognition of the five genera previously synonymized with *Notropis* (*Alburnops*, *Hudsonius, Hydrophlox, Miniellus* and *Paranotropis*) and re-allocated some species within these groups with other genera within

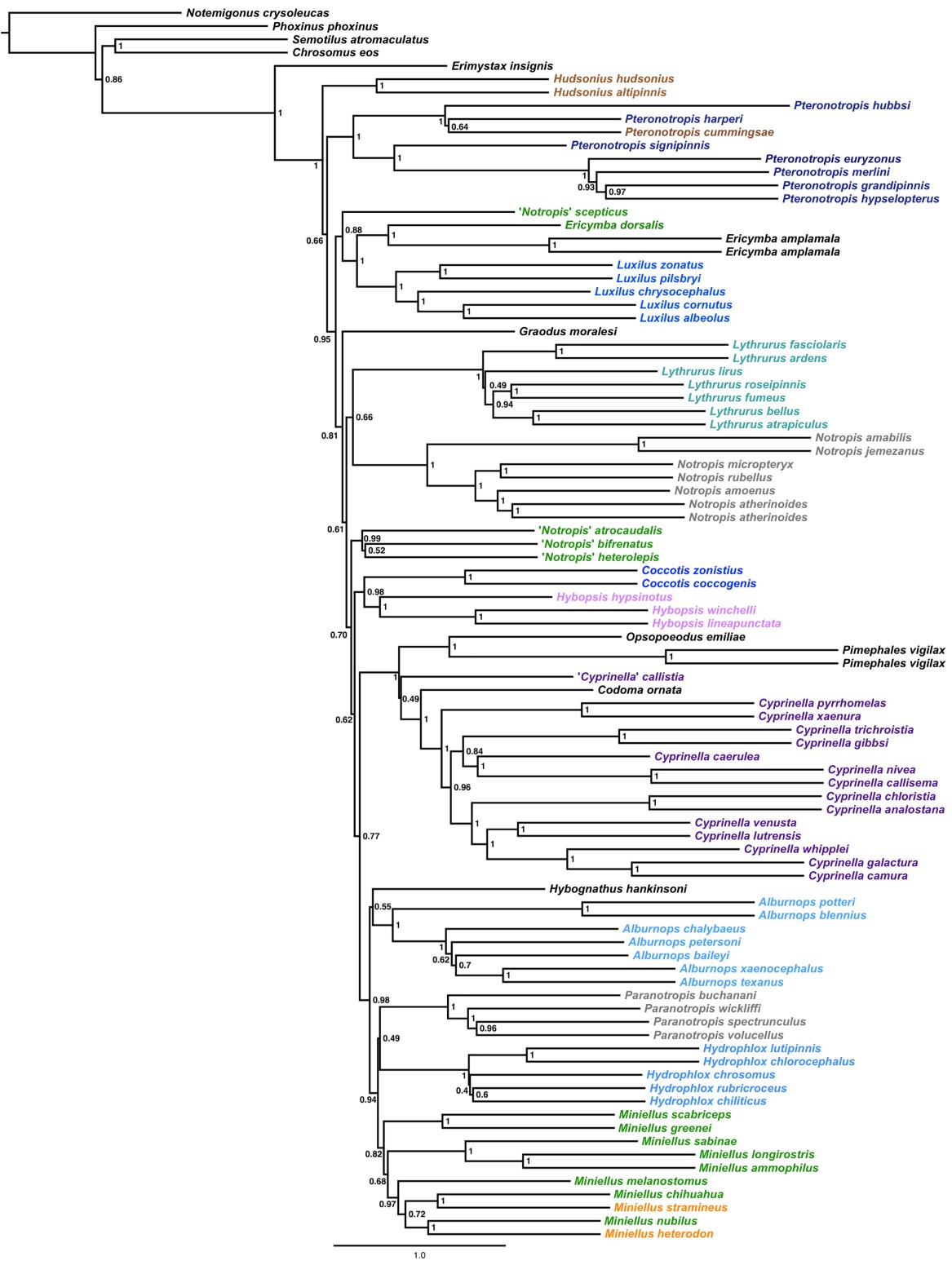

**Figure 3 Species tree using ASTRAL-II.** Internal branch lengths are in coalescent units and branches that lead to tips are not calculated by ASTRAL-II but instead arbitrarily displayed. Branch support values indicate the support for a quadripartition (instead of bipartitions).

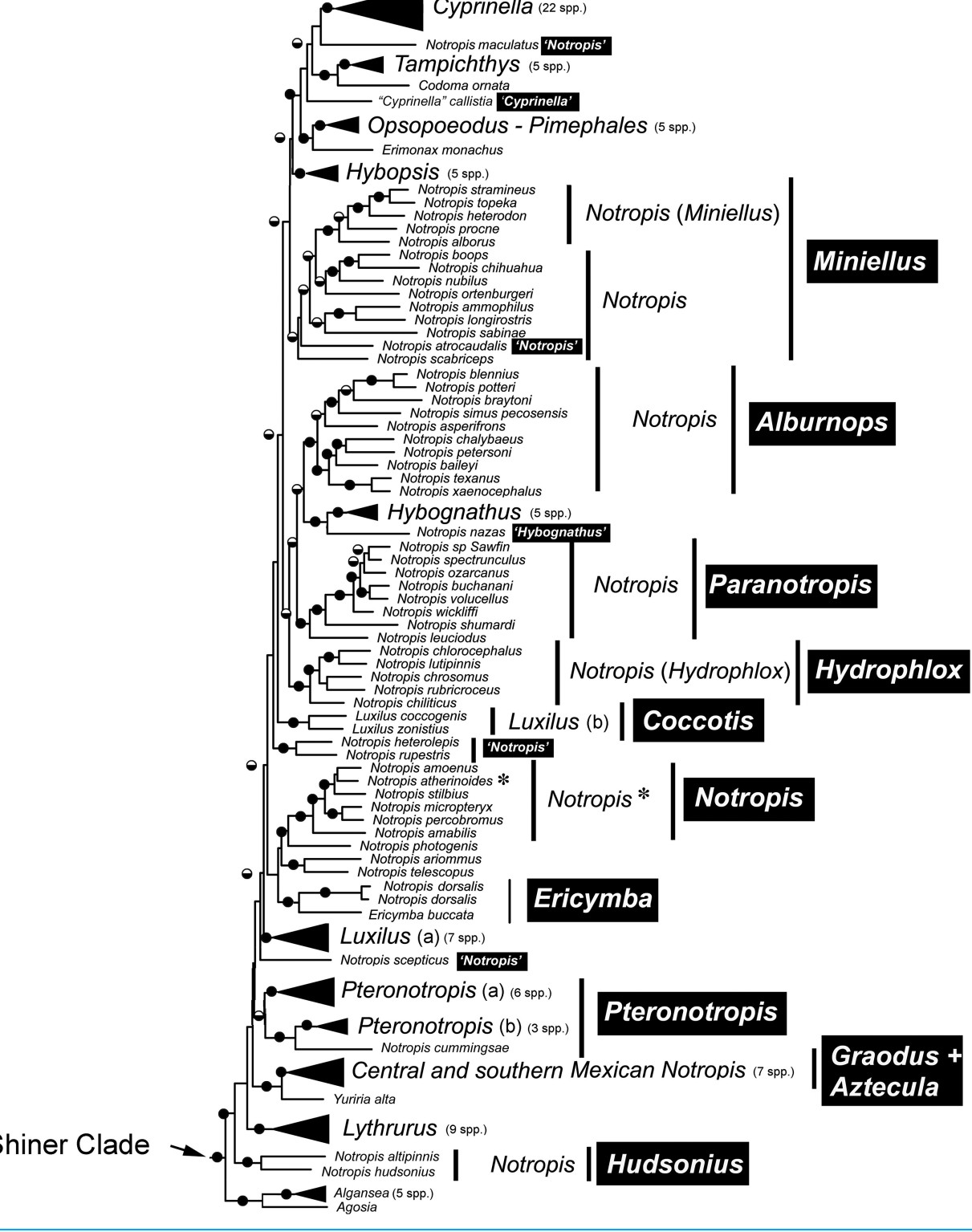

**Figure 4  Major monophyletic groups and genera within the Shiner Clade based on two nuclear and two mitochondrial genes (modified from _Schönhuth et al., 2018_).** Black boxes with white text represent name changes. Circles at nodes represent >75% bootstrap values (BS, black top) and >95% Baysian posterior probability (PP, black bottom).

the Shiner Clade that differed from prior studies (*Mayden, 1989*; *Mayden et al., 2006*; *Gidmark & Simons, 2014*). This study also highlights some interesting relationships with other closely related genera within the Shiner Clade.

### *Pteronotropis* and *Hudsonius*

All three species of *Hudsonius* were included in the analysis but were not recovered as monophyletic, with *H. cummingsae* grouping within *Pteronotropis*. The range for all three species of *Hudsonius* overlaps with that of *Pteronotropis* across the southeastern states of North and South Carolina, Georgia, Alabama, and Florida, but only *H. hudsonius* extends northward up through the Great Lakes and across much of Canada. *Mayden et al. (2006)* found support for a monophyletic *Hudsonius*, despite individuals of *H. altipinnis* not forming a lineage, suggesting cryptic speciation. *Hollingsworth et al. (2013)* found *Hudsonius* as monophyletic in concatenated analyses and with cytochrome *b*, but within *Pteronotropis* with just the nuclear gene Rag1. In our analysis, *Hudsonius cummingsae* is part of *Pteronotropis* (BS 100, PP 1), while *H. altipinnis* (collected in South Carolina) and *H. hudsonius* (collected in Wisconsin) were found as sister species (BS 100, PP 1), congruent with *Schönhuth et al. (2018*; Fig. 4). Given that mitochondrial markers are suggesting the monophyly of *Hudsonius* and nuclear markers are suggesting that *H. cummingsae* is within *Pteronotropis*, it is possible that there has been mitochondrial introgression between *H. cummingsae* and *H. altipinnis*; however, this will need further examination. Because *H. hudsonius* is the type species, we propose removing *Hudsonius cummingsae* from *Hudsonius* and transferring it to *Pteronotropis* to maintain monophyly of both genera *Pteronotropis* and *Hudsonius*. *Pteronotropis* is, in part, diagnosed by a wide, dark stripe on the body, a character that *P. cummingsae* shares.

*Mayden (1989)* and *Simons, Knott & Mayden (2000)* did not find *Pteronotropis* to be monophyletic; however the genus was monophyletic here (BS 100, PP 1) as well as in *Mayden & Allen (2015)* and *Schönhuth et al. (2018)*. There are two deeply divergent clades, that of *P. harperi* + *P. hubbsi* + *P. cummingsae* + *P. welaka*, and that of *P. euryzonus* + *P. grandipinnis* + *P. hypelopterus* + *P. merlini* + *P. metallicus* + *P. signipinnis* + *P. stonei*, and it could be argued that these two clades deserve separate genus status. This deep divergence may have led to nonmonophyly in earlier studies without either enough characters and/or taxa in the analyses. In *Schönhuth et al. (2018)*, the monophyly of *Pteronotropis* was supported only under Bayesian inference and not Maximum Likelihood (Fig. 4).

### *Luxilus*

Our analysis includes seven of the nine recognized species of *Luxilus* and recovers two distinct clades. *Luxilus chrysocephalus* (type species) forms a clade (BS 100, PP 1) with *L. zonatus*, *L. pilsbryi*, *L. albeolus*, and *L. cornutus* that is sister to *Ericymba* + 'Notropis' *dorsalis*. Two other species, *Luxilus coccogenis* and *L. zonistius*, are resolved as a clade (BS 100, PP 1) distant to other members of *Luxilus* and instead sister to *Hybopsis*. *Mayden (1989)* removed *Luxilus* from *Notropis*, considering it sister to *Cyprinella* (including *Codoma ornata*) and monophyletic based on three morphological characters, while

*Coburn & Cavender (1992)* considered *Luxilus* sister to a clade comprised of *Lythrurus*, *Cyprinella*, *Pimephales*, and *Opsopoeodus*, and they noted *Luxilus* could be an unnatural assemblage. Morphological and molecular studies primarily focused on members within *Luxilus* have assumed the monophyly of the genus, rather than including other genera, and have consistently found a sister relationship between *L. coccogenis* + *L. zonistius*, supporting our findings (*Gilbert, 1964*; *Menzel, 1976*; *Dowling et al., 1992*; *Dowling & Naylor, 1997*; *Mayden et al., 2006*); although an allozyme analysis found *L. coccogenis* and *L. zonistius* to be sister to *L. cerasinus* (*Buth, 1979*). Other studies that included *Luxilus* and a variety of other shiner taxa have argued that *Luxilus* is not monophyletic (*Simons, Berendzen & Mayden, 2003*; *Mayden et al., 2006*; *Schönhuth & Mayden, 2010*; *Hollingsworth et al., 2013*). We support that *L. coccogenis* + *L. zonistius* should no longer be included in *Luxilus*, and propose elevating the genus *Coccotis* Jordan 1882 (*Coccotis coccogenis* Jordan 1882 as the type species) for these taxa. These results are also in agreement with *Schönhuth et al. (2018*; Fig. 4) where *Luxilus* was resolved as nonmonophyletic.

### Lythrurus

*Lythrurus* has long been considered monophyletic (*Snelson, 1972*; *Schmidt, Bielawski & Gold, 1998*; *Mayden et al., 2006*; *Pramuk et al., 2007*), and our study also supports the monophyly of this group (BS 100, PP 1). What has been more problematic, however, is determining the clade's relationship to other genera. Formerly considered sister to a *Luxilus* + *Cyprinella* clade (*Mayden, 1989*), it was later poorly resolved by *Mayden et al. (2006)* in a clade with various 'Notropis' species. *Coburn & Cavender (1992)* determined *Lythrurus* was sister to a clade comprised of *Cyprinella*, *Pimephales*, and *Opsopoeodus*, and more recently it has been weakly resolved as the sister group to all other species of the Shiner Clade excluding *Algansea-Agosia*, and *Hudsonius* (*Schönhuth et al., 2018*). We find moderate support (BS 90, PP 0.66) for *Lythrurus* as sister to true *Notropis* (the clade containing *Notropis atherinoides*, the type species of *Notropis*; more discussion on *Notropis* below).

### Cyprinella

Although *Gibbs (1957)* and *Mayden (1989)* found *Cyprinella callistia* as nested within *Cyprinella*, *Broughton & Gold (2000)* found the species to be sister to the remaining species of *Cyprinella* (*Codoma* and *Tampichthys* were not included). One of the most extensive and recent molecular studies of *Cyprinella* and relatives (*Schönhuth & Mayden, 2010*) failed to resolve the genus as monophyletic, relative to *C. callistia*. The remainder of *Cyprinella* was monophyletic, and was sister to *Codoma* + *Tampichthys*; however, *Cyprinella callistia* was either sister to the clade of *Cyprinella* + *Codoma* + *Tampichthys* or as sister to that clade plus *Pimephales* + *Opsopoeodus*. We did not include *Tampichthys*, but we also resolved *Codoma* more closely related to all other representatives of *Cyprinella* (BS 100, PP 1) than *Cyprinella callistia*. As in previous analyses, we could not resolve the node

leading to *Cyprinella callistia*, *Opsopoeodus* + *Pimephales*, and *Codoma* + *Cyprinella* (BS < 70, PP 0.49), but we clearly show (based on topology and the AU test) that *Cyprinella callistia* should not be included in *Cyprinella* (*Schönhuth & Mayden, 2010*; *Schönhuth et al., 2018*). *Cyprinella callistia* was originally described as *Photogenis callistius* (Jordan 1877), but this genus does not apply to *C. callistia*. We did not include the type of the genus, *Photogenis photogenis* (*Notropis photogenis*), in our analysis, but *N. photogenis* has been resolved within the clade including the true *Notropis*, and was not closely related to *C. callistia* (*Schönhuth et al., 2018*). With no name available for the species, we refer to it as '*Cyprinella*' *callistia* until such time that a broader analysis can be completed. This name will reflect that this species is clearly divergent from other *Cyprinella*, both morphologically (*Mayden, 1989*) and genetically (*Schönhuth & Mayden, 2010*; *Schönhuth et al., 2018*; this study).

We did not examine *Notropis maculatus*; however, *Schönhuth et al. (2018)* found it to be sister to *Cyprinella*. *Notropis maculatus* is found in muddy coastal streams, backwaters, and oxbows along the Gulf and Atlantic coasts. Like *Cyprinella*, it has instense pigmentation in nuptial males and has broad scales outlined in black. In *Schönhuth et al. (2018)*, its position in the phylogeny was well supported only under Bayesian analysis, and further research is needed to determine its position, and we recognize it as '*Notropis*' *maculatus*.

### Alburnops and Hydrophlox

*Gidmark & Simons (2014)* resurrected *Alburnops* based on the monophyly recovered by *Mayden et al. (2006)*. We do not recover monophyly of the *Alburnops* species here analyzed, and instead find primarily two non-sister clades. The type species, *Alburnops blennius*, is recovered in a clade with *A. baileyi*, *A. chalybaeus*, *A. petersoni*, *A. potteri*, *A. texanus*, and *A. xaenocephalus* (BS 100, PP 1), and thus these should retain the genus name. *Schönhuth et al. (2018*; Fig. 4) also recovered this well supported clade including these seven species plus *A. asperifrons*, *A. braytoni*, and *A. simus pecosesis*. Additionally, *A. bairdi*, *A. buccula*, *A. candidus*, *A. edwardraneyi*, *A. girardi*, *A. hypsilepis*, and *A. shumardi* should be included in *Alburnops* (*Mayden, 1989*; *Mayden et al., 2006*; *Gidmark & Simons, 2014*). *Notropis aguirrepequenoi* was described from out of *A. braytoni* and is included here in *Alburnops* (*Miller, Minckley & Norris, 2005*). *Notropis orca* is likely extinct, but similar to *A. simus* and is included in *Alburnops* (*Chernoff & Miller, 1986*; *Mayden, 1989*), but this will need to be confirmed.

The other clade (BS 100, PP 1) composed of species included within *Alburnops* by *Gidmark & Simons (2014)* includes *A. chiliticus*, *A. chlorocephalus*, *A. chrosomus*, *A. lutipinnis*, and *A. rubricroceus*, the five species recognized as *Hydrophlox* by *Cashner, Piller & Bart (2011*; type species *Hybopsis rubricroceus* Cope). This clade is here more closely related to species currently recognized under *Notropis*, '*Notropis*', and *Miniellus* than to the *Alburnops* clade. Our results agreed with recent phylogenetic analyses by *Schönhuth et al. (2018*; Fig. 4) that also differentiated this clade of five species, and we recognize *Hydrophlox* as valid.

### Miniellus + some 'Notropis'

*Miniellus* is currently recognized as containing four species: *Miniellus procne* (type species), *M. heterodon*, *M. stramineus*, and *M. topeka* (*Mayden et al., 2006*; *Gidmark & Simons, 2014*). While our analyses did not include *M. procne* or *M. topeka*, *M. stramineus* was not resolved sister to *M. heterodon*. Instead, several 'Notropis' species were found to be more closely related to these two species. Five of these 'Notropis' were considered by *Mayden et al. (2006)* as belonging to a 'Notropis' longirostris clade. *Schönhuth et al. (2018*; Fig. 4) found strong support for a monophyletic *Miniellus* including the four initial species (*Miniellus procne*, *M. heterodon*, *M. stramineus*, and *M. topeka*) sister to 'N.' alborus (not analyzed here). Both studies also included in this clade 'N.' sabinae, 'N.' longirostris, 'N.' ammophilus, 'N.' chihuahua, 'N.' nubilus and 'N.' scabriceps with *Schönhuth et al. (2018)* also including 'N.' ortenburgeri, and 'N.' boops, and the present study also including 'N.' greenei and 'N.' melanostomus. However, while both studies analyzed 'N.' atrocaudalis, in *Schönhuth et al. (2018)* this species was collapsed with 'N.' scabriceps as sister to the rest of species in this clade; and in the present study 'N.' atrocaudalis was not resolved within this clade but in a clade with 'N.' bifrenatus and 'N.' heterolepis. Given strong support in the present study for the clade (BS 93, PP 0.82) that includes *Miniellus*, the 'N.' longirostris clade, and these other species of 'Notropis', we expand the genus *Miniellus* to include 'Notropis' greenei, 'N.' scabriceps, 'N.' sabinae, 'N.' longirostris, 'N.' ammophilus, 'N.' chihuahua, 'N.' melanostomus, and 'N.' nubilus, plus three species not included in the present analysis ('N.' alborus, 'N.' ortenburgeri, and 'N.' boops) that were also resolved within this clade (*Schönhuth et al., 2018*). Additionally, 'N.' mekistocholas and 'N.' rafinesquei, should be considered within this extended *Miniellus*, based on original descriptions (*Snelson, 1971*; *Suttkus, 1991*) and their strongly supported position as part of the 'Notropis' longirostris clade (*Suttkus & Boschung, 1990*; *Mayden et al., 2006*). 'Notropis' albizonatus and 'N.' uranoscopus are included in *Miniellus* per *Warren, Burr & Grady (1994)*. *Notropis perpallidus* was found in *Hollingsworth et al. (2013)* to be sister to *Miniellus sensu stricto* and *N. anogenus* was sister to *N. ortenburgeri*, so both are included in *Miniellus* here. We note that further investigation needs to be done to resolve relationships within this clade.

### 'Notropis'

Besides the species listed above that group with *Miniellus*, several other 'Notropis' are found throughout our phylogeny. 'Notropis' scepticus is found sister to *Hudsonius* in the concatenated analysis (BS 97) but sister to *Ericymba* + *Luxilus* in the species tree (PP 0.88). Sister relationship of this divergent species was also unresolved within the Shiner Clade in *Schönhuth et al. (2018)*. The phylogenetic position of 'N.' scepticus varies in different studies, and despite the fact that we retain it under 'Notropis', the best solution nay be to describe a separate genus for this species.

We do not recover the 'Notropis' dorsalis group (*Mayden, 1989*; *Raley, Wood & McEachran, 2001*) as monophyletic. This group was composed of 'Notropis' dorsalis, 'N.' ammophilus, 'N.' longirostris, 'N.' rafinesquei, and 'N.' sabinae. Instead we find 'N.' ammophilus, 'N.' longirostris, and 'N.' sabinae grouping with *Miniellus* (see above), while

'N.' dorsalis was strongly supported as sister to *Ericymba* (BS 100, PP 1), as in *Schönhuth et al. (2018)*. Currently, *Ericymba* is diagnosed by the presence of enlarged infraorbital canal scales (*Pera & Armbruster, 2006*), which are not found in 'N.' dorsalis; however, 'N.' dorsalis is otherwise very similar in morphology to the species of *Ericymba*, having a large mouth and ventrally flattened body. Given that 'N.' dorsalis is strongly resolved sister to *Ericymba* we are recognizing the species as *Ericymba dorsalis*; however, the species should be examined in greater detail to detemine if it requires a separate genus.

A clade of 'Notropis' heterolepis, 'N.' bifrenatus, and 'N.' atrocaudalis was strongly supported (BS 100, PP 0.99) with the clade sister to *Ericymba + Luxilus* in the concatenated analysis (BS 76) and *Coccotis, Hybopsis, Opsopoeodus, Pimephales*, 'Cyprinella' callistia, *Codoma, Cyprinella, Hybognathus, Alburnops, Paranotropis, Hydrophlox*, and *Miniellus* in the species tree (PP 0.70). These relationships differ signficantly with those of *Schönhuth et al. (2018)* where 'N.' heterolepis was resolved sister to 'N.' rupestris (a morphologically similar species not analyzed here) in a well supported and divergent clade with undefined relationships, 'N.' atrocaudalis was sister to the remaining extended *Miniellus* clade minus 'N.' scabriceps, and 'N.' bifrenatus was not examined. Recognizing the differences between the studies, we retain the following species as 'Notropis': 'N.' atrocaudalis, 'N.' bifrenatus, 'N.' heterolepis, and 'N.' rupestris.

The relationships of two small species from northeastern Mexico, 'Notropis' saladonis 'N.' tropicus have not been examined. 'Notropis' saladonis is currently considered to be extinct (*Mercado Silva, 2019*).

### Notropis sensu stricto and Paranotropis

We find two clades that include species regarded as true *Notropis* (*Mayden et al., 2006*; *Gidmark & Simons, 2014*). The type species, *Notropis atherinoides*, is found in a clade (BS 100, PP 1) that is sister to *Lythrurus* (BS 90, PP 0.66) and contains *N. amabilis, N. amoenus, N. jemezanus, N. micropteryx*, and *N. rubellus*. Whithin this clade, *Schönhuth et al. (2018)* also included *N. percobromus, N. photogenis*, and *N. stilbius* and *Hollingsworth et al. (2013)* also included *N. oxyrhynchus* and *N. suttkusi*. *Notropis* should be limited to just these species. *Schönhuth et al. (2018)* weakly supported the clade of *N. ariommus* and *N. telescopus* to *Notropis sensu stricto*, and these relationships should be further studied, thus we refer to the species as 'N.' ariommus and 'N.' telescopus as well as the likely relative 'N.' semperasper (*Coburn, 1982*).

The other clade includes *N. buchanani, N. wickliffi, N. volucellus*, and *N. spectrunculus* (BS 100, PP 1), and here this clade forms a polytomy with the *Hydrophlox* and *Miniellus* clades in the concatenated analysis (BS 100, PP 1) and poorly supported as the sister to *Hydrophlox* in the species tree analysis (pp 0.49). *Schönhuth et al. (2018)* also included *N. leuciodus, N. ozarcanus* and *N. shumardi*, within this well supported clade. These species were either originally described as *Notropis*, or have been moved to *Notropis* from *Alburnops, Hybognathus*, or *Hybopsis*. *Notropis leucidous* is the type species of *Paranotropis* Fowler 1904, and its placement within the clade is well supported (*Simons, Berendzen & Mayden, 2003*; *Schönhuth et al., 2018*); thus, we propose these species be

considered as *Paranotropis*. Also included is *N. cahabae* per the original description (*Mayden & Kuhajda, 1989*).

## Central and southern Mexican *Notropis*

There were several species, mostly from Mexico, that were not examined in this study that were recognized within a clade called "Central and southern Mexican *Notropis*" (CSMN clade) in *Schönhuth et al. (2018*: 788). The CSMN clade contains species that have been placed into three genera: *Aztecula*, *Graodus*, and *Yuriria*. Of the species in the CSMN clade, we only examined *G. moralesi*.

*Schönhuth et al. (2018)* examined four species of *Aztecula* (listed as *N. sallaei* - type species, *N. calientis*, *N. grandis*, and *N. marhabatiensis*), three *Graodus* (listed as *N. boucardi*, *N. imeldae*, and *N. moralesi*), and one *Yuriria* (*Y. alta*). *Aztecula* was not monophyletic with *A. sallaei* sister to the species of *Graodus*; however, in previous studies (*Schönhuth & Doadrio, 2003*; *Schönhuth, Doadrio & Mayden, 2006*; *Schönhuth et al., 2008*; *Hollingsworth et al., 2013*), *Aztecula* and *Graodus* were both monophyletic.

*Aztecula sallaei* has a complex taxonomic history that includes many synonyms, movement between many genera, and a temporary change in spelling of its specific epithet to *sallei* (*Chernoff & Miller, 1981*). *Aztecula* additionally includes species that were described from out of *N. calientis*: *N. amecae*, *N aulidion*, *N. calabazas*, *N. grandis*, and *N. marhabatiensis* (*Chernoff & Miller, 1986*; *Lyons & Mercado-Silva, 2004*; *Domínguez-Domínguez et al., 2009*). The type species of *Graodus* is *G. nigrotaeniatus* Günther, 1868, which is believed to be a synonym of *G. boucardi* (*Miller, Minckley & Norris, 2005*). *Graodus* additionally includes *N. cumingii*, which is considered by some to be a senior synonym of *G. imeldae* (*Gilbert, 1998*; *Miller, Minckley & Norris, 2005*; *Page et al., 2013*) and an undescribed species in Oaxaca (referred as *Notropis* sp. 1 in *Schönhuth, Doadrio & Mayden, 2006*; *Schönhuth et al., 2008*). The species of *Graodus* and *Aztecula* (with the exception of *A. sallaei*) were considered to be in the genus *Hybopsis* (*Miller, Minckley & Norris, 2005*); however, they are not closely related to *Hybopsis* in any phylogenetic study. *Yuriria* was considered valid with the additional species *Y. chaplalae* (*Miller, Minckley & Norris, 2005*). A third species, *Y. amatlana*, has been described (*Domínguez-Domínguez, Pompa-Domínguez & Doadrio, 2007*). The species and genera of the CSMN clade will need further review and morphological examination, but given the strong support for these clades in previous studies (*Schönhuth, Doadrio & Mayden, 2006*; *Schönhuth et al., 2008*, *2018*), we retain the taxonomy per *Gidmark & Simons (2014)* in three genera (*Graodus*, *Aztecula* and *Yuriria*) with some additional species placed as above.

## Differences with *Schönhuth et al. (2018)*

Certain species/clades within the Shiner Clade have inconsistent phylogenetic placement between the present and *Schönhuth et al. (2018)* analyses, or were not resolved within any of these groups in either of these studies. These species include 'Cyprinella' callistia, 'Notropis' scepticus, 'N.' heterolepis + 'N.' rupestris, 'N.' bifrenatus, 'N. ariomus', 'N.' telescopus, 'N.' nazas, and 'N.' maculatus. In *Schönhuth et al. (2018)* Notropis nazas was strongly resolved sister to *Hybognathus*, and we suggest it to be considered as

'*Hybognathus*' until it can be further validated as a species of *Hybognathus* or as a separate genus. All these other taxa should be futher examined with genomic and morphological data with taxon sampling equal to or greater than that of *Schönhuth et al. (2018)*; until that time descriptions of new genera or elevations of old genera for these species is premature.

### Intraspecies utility of *Arcila et al. (2017)* markers

This study included two specimens of three species: *Notropis atherinoides, Ericymba amplamala*, and *Pimephales vigilax*. *Notropis atherinoides* specimens, one from Wisconsin and the other from Arkansas (both Mississippi River drainage), exhibited 99.6% sequence similarity with a pairwise distance of 0.003 and a total of 1,086 nucleotide differences across the entire 286,455 bp alignment. The specimens of *E. amplamala* were from Alabama (Mobile River Drainage) and Mississippi (Pascagoula River drainage), populations that were not found to be morphologically distinguishable in a detailed analysis (*Pera & Armbruster, 2006*), and had 99.5% sequence similarity, a pairwise distance of 0.005, and 1,424 differences. Our samples of *Pimephales vigilax* were collected from Paint Rock River (Tennessee River Drainage) and Uphapee Creek (Mobile River Drainage) and had 99.7% sequence similarity, a pairwise distance of 0.002, and 845 nucleotide differences. These results suggest two things: there may be cryptic diversity within shiner clade species, and the *Arcila et al. (2017)* markers are likely of utility at the population level, despite their initial development for use across a very broad taxonomic scale. *Rincon-Sandoval, Betancur-R & Maldonado-Ocampo (2019)* also found the markers useful for elucidating phylogeographic patterns within species in the neotropics and that they may have better utility than nuclear markers developed from a RADseq approach.

One of the targeted sequences was COI, a popular mitochondrial marker that is often used to delineate fish species and that can provide a comparison with the phylogenomic markers as a whole. We find a wide range of infraspecific differences in the 703 bp of the partial COI sequences examined. *Notropis atherinoides*, despite distant collection sites, has only a 2 bp difference (0.4% divergence). *Pimephales vigilax* from the neighboring Tennessee and Mobile River systems had a 16 bp difference (2.3% divergence). *Ericymba amplamala*, however, had a 54 bp difference (7.7% divergence), a degree of difference often associated with species-level differentiation, and there needs to be further investigation into the genetic structure of the species. COI alone may be suitable for identification of cryptic diversity for shiners, but the full phylogenomic dataset adds a considerable number of characters for elucidating population structure.

## CONCLUSIONS

This study provides an important first step in using phylogenomics to resolve the relationships and taxonomy of the problematic leuciscid minnows. By employing a publicly available probe set (*Arcila et al., 2017*), future research can include more specimens that were not sampled in this study and easily be combined with our dataset. Our phylogenies help in understanding why this group has been difficult to resolve. A phylogenomic approach provides far more characters that can break the polytomies at the base of the shiner clade that are the likely result of rapid divergence. Not only has the group

been described as morphologically conserved (*Gidmark & Simons, 2014*), thus hampering morphological interpretations of relationships, but we would argue that the same is true genetically. We find over 88% similarity (or uninformativeness) in a dataset comprised of over 288,000 base pairs. However, we find strong agreement between this study and the four gene phylogeny provided by *Schönhuth et al. (2018)*, indicating that dense taxonomic sampling, as done in the latter study, is also a key in resolving closely related taxa. Problems with elucidating shiner relationships have been exacerbated by studies focusing only on subsets of the shiner clade due to sampling or cost restrictions. We demonstrate the utility of the exon capture method of *Arcila et al. (2017)* to elucidate relationships of rapidly evolving clades, and demonstrate that the markers may be of use at the population level as well. This is important as studies utilizing the *Arcila et al. (2017)* markers have the potential of resolving deep and shallow relationships within a single analaysis (*Hughes et al., 2021*). With the continuing decrease in cost of phylogenomic methods, the demonstrable utility of the *Arcila et al. (2017)* markers at many phylogenetic levels, and the soon to be large number of fish taxa sampled using the *Arcila et al. (2017)* markers, we would encourage researchers to add to this dataset. Numerous issues remain in the taxonomy and systematics of North American leuciscids, and we will continue to add species to the analysis. This study continues the trend at subtending the shiner clade into genera, but several important clades still need to be resolved and described.

## ACKNOWLEDGEMENTS

We thank Dr. Kyle Piller for providing tissues to supplement those provided by the Auburn University Museum of Natural History and the St. Louis University Fish Collection. Dr. David Neely, Alan Cressler, and ncfishes.com generously provided permission for use of copyrighted fish images; some fish images were available *via* public domain. This article is contribution No. 953 of the Auburn University Museum of Natural History.

### Funding

This work was supported by the United States National Science Foundation "All Cypriniformes Species Inventory" Grant (DEB-1023403 to Jonathan W. Armbruster and DEB-1021840 to Richard L. Mayden). The funders had no role in study design, data collection and analysis, decision to publish, or preparation of the manuscript.

### Grant Disclosures

The following grant information was disclosed by the authors:
United States National Science Foundation "All Cypriniformes Species Inventory": DEB-1023403, DEB-1021840.

### Competing Interests

The authors declare that they have no competing interests.

## Author Contributions

- Carla Stout conceived and designed the experiments, performed the experiments, analyzed the data, prepared figures and/or tables, authored or reviewed drafts of the article, and approved the final draft.
- Susana Schonhuth analyzed the data, prepared figures and/or tables, authored or reviewed drafts of the article, and approved the final draft.
- Richard Mayden analyzed the data, prepared figures and/or tables, authored or reviewed drafts of the article, and approved the final draft.
- Nicole L. Garrison performed the experiments, analyzed the data, authored or reviewed drafts of the article, and approved the final draft.
- Jonathan W. Armbruster conceived and designed the experiments, analyzed the data, prepared figures and/or tables, authored or reviewed drafts of the article, and approved the final draft.

## DNA Deposition

The following information was supplied regarding the deposition of DNA sequences:

The data is available in the Supplemental File and at GenBank: PRJNA842507.

## Data Availability

The alignment and the partition are available in the Supplemental File.

## Supplemental Information

Supplemental information for this article can be found online at http://dx.doi.org/10.7717/peerj.14072#supplemental-information.

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
