# Peer review of "Phylogenomics and classification of Notropis and related shiners (Cypriniformes: Leuciscidae) and the utility of exon capture on lower taxonomic groups"

_PeerJ, doi:10.7717/peerj.14072_

## Round 0.1 · original submission · Minor Revisions

We have received two reviews for your study and, as you will read below, both reviewers were very positive about it.

Nonetheless, there are a few issues that deserve to be revised. In particular, reviewer 1 identified some aspects of the methods that should be clarified and pointed out that the criteria used for reclassifying genera should be more detailed. Reviewer 2 also suggested additional discussion and relevant references to further improve the manuscript. Finally, the description (of legends) and resolution of figures need to be improved.

Reviewer 1 ·

Basic reporting

In general, I find the manuscript well-written. I could not find an explicit data deposition statement in the manuscript, although I do see that SRA numbers are listed as pending in the Materials and Methods. I think it would be better if there was a separate statement with these numbers, and a brief description of what is in the online supplement.

I do have a few comments about the figures. In general, I thought that the figure legends were a little sparse and could have used more information.

Figures 2-3: In Fig. 2, the authors use their new generic classification for the terminal names (i.e. Paranotropis, expanded Miniellus). In Fig. 3, they use the old classification. I find this confusing. It is also not stated what the different colors on the tip labels mean (if anything)?

Figure 4: What do the circles and half circles on the nodes represent in this figure? The legend doesn’t say. I don’t quite understand what the red boxes represent, could the authors clarify in the legend? Would the authors consider using a second color for the new classification, so these two things are easy to differentiate?

Experimental design

There are a few pieces of information I think the authors need to add to the methods for clarity:

Line 156: What Illumina platform was used for sequencing (HiSeq 2500, HiSeq 2000, NextSeq, etc)? Was the sequencing paired-end or single-end?

Line 168: Did the authors use the Geneious aligner, or another software within the Geneious ecosystem (i.e. Muscle, Mafft, etc)?

Line 172: Can the authors be more specific about how the loci were partitioned?

Line 182: Here and in the abstract, the authors mention that ~11% of sites were phylogenetically informative, but it is not clear how this was calculated and does not appear in the methods. It also doesn't really come up again in the discussion. Is this parsimony informative sites? Or some other metric of phylogenetic informativeness?

Validity of the findings

Lines 219-221: Did the authors choose a specific support cutoff to reclassify these shiner clades? For example, only updating genera for clades with 100% support in both analyses? It seems like that’s mostly the case looking at Figures 2-3, except maybe with Miniellus. There is also a clade of ‘Notropis heterolepis’, ‘Notropis bifrenatus’, and ‘Notropis atrocaudalis’ that the authors did not reclassify. I would just like the authors to be a little more explicit with their criteria for reclassifying genera based on their phylogenomic trees.

Reviewer 2 ·

Basic reporting

Language - This paper is well-written in clear, professional English, with no evident grammatical issues.

References -
The authors have included many relevant recent works in the references section, but failed to include some foundational literature on Notropis phylogeny and evolution.

Structure - Additional discussion is warranted when explicitly comparing the results of this study with Schonhuth 2018. Some details about branching order and node support were not discussed.

Figures - Inconsistent image resolution across figures. Particularly, Figure 2 is hardly legible in the pdf distributed to reviewers.

Self-contained with relevant results - I have spent many hours looking for ways to improve this manuscript, but have failed. Although I would have liked to see higher taxonomic coverage, the authors did include representatives from each of the known clades within "Notropis." I encourage the authors to build upon the dataset to include additional taxa, especially those of conservation concern. After the editor addresses the figure resolution issues, this manuscript is ready for immediate publication.

Experimental design

See comment above regarding self-contained, relevant results. The taxonomic coverage is less extensive than previous research, but the experimental design is well-defined and meaningful. The inclusion of species trees provides critical validation of results from concatenated data. This is a necessary step for all multi-locus analyses, and the authors explained their approach in detail.

Validity of the findings

This paper fairly represents the

Additional comments

Bravo!

---

## Round 0.2 · accepted · Accept

I am generally satisfied with the final revisions made to the manuscript.